# Hepatitis B Virus Infection: A Mini Review

**DOI:** 10.3390/v16050724

**Published:** 2024-05-03

**Authors:** Diana Asema Asandem, Selorm Philip Segbefia, Kwadwo Asamoah Kusi, Joseph Humphrey Kofi Bonney

**Affiliations:** 1West African Center for Cell Biology of Infectious Pathogens, University of Ghana, Accra P.O. Box LG 52, Ghana; dasandem@noguchi.ug.edu.gh; 2Department of Virology, Noguchi Memorial Institute for Medical Research, University of Ghana, Accra P.O. Box LG 581, Ghana; 3Department of Immunology, Noguchi Memorial Noguchi Memorial Institute for Medical Research, University of Ghana, Accra P.O. Box LG 581, Ghana; ssegbefia@noguchi.ug.edu.gh (S.P.S.); akusi@noguchi.ug.edu.gh (K.A.K.)

**Keywords:** viral hepatitis, Hepatitis B virus, Hepatitis D virus, Human Immunodeficiency Virus, vaccination, Ghana, liver cancer, hepatocellular carcinoma, liver coinfections, *Plasmodium* infections

## Abstract

Hepatitis B and C viruses (HBV and HCV) are the leading causes of end-stage liver disease worldwide. Although there is a potent vaccine against HBV, many new infections are recorded annually, especially in poorly resourced places which have lax vaccination policies. Again, as HBV has no cure and chronic infection is lifelong, vaccines cannot help those already infected. Studies to thoroughly understand the HBV biology and pathogenesis are limited, leaving much yet to be understood about the genomic features and their role in establishing and maintaining infection. The current knowledge of the impact on disease progression and response to treatment, especially in hyperendemic regions, is inadequate. This calls for in-depth studies on viral biology, mainly for the purposes of coming up with better management strategies for infected people and more effective preventative measures for others. This information could also point us in the direction of a cure. Here, we discuss the progress made in understanding the genomic basis of viral activities leading to the complex interplay of the virus and the host, which determines the outcome of HBV infection as well as the impact of coinfections.

## 1. Introduction

The liver is an important multifunctional organ in the body and plays various metabolic and immunologic roles, making its malfunctioning critically fatal. Though there are varied therapeutic strategies, one of the best options for treatment in terminal liver infections is liver transplantation. This in itself is limited by restricted organ supply, the complexity of human leukocyte antigen (HLA) matching, immunologic side effects, and financial constraints [1]. Moreover, long-term survival with this option is not guaranteed.

Dysregulation of liver inflammation is a hallmark of chronic infection, autoimmunity, and malignancy, which is mediated by multiple overlapping pathways in different liver diseases. Immune dysregulation can also be targeted as a therapeutic strategy in liver disease. Certain liver diseases are characterised by dysregulated immune function, where the body’s attempt to eliminate the source of insult is either exacerbated beyond control or prolonged beyond containment [2,3]. Thus, a ‘collateral damage’ phenomenon ensues where the body’s aggravated defence mechanism tends to destroy self-cells. In some cases, the response may not necessarily be aggravated but rather prolonged and persistent, leading to destruction of hepatocytes and development of scar tissue. The body’s repair mechanisms may not be adequate to keep up with the constant damage of hepatocytes.

The term hepatitis is used to describe the totality of liver inflammation [4]. Although liver damage may be a result of over-exertion of the immune system, it could be triggered by a foreign agent commonly of viral aetiology [5]. Viral components and life processes such as replication activate an immune response that could result in an acute or chronic infection [6]. This is dependent on the infecting pathogen as well as the host and results in various outcomes.

The most common cause of viral hepatitis is Hepatitis B virus (HBV), which is a leading cause of end-stage liver disease worldwide [7]. Although there is an effective vaccine against HBV infection, new infections continue to occur due partly to limited vaccination coverage and the accessibility, availability, and cost of the vaccines in the highest-burdened regions [8]. Another avenue for new infections stems from viral breakthrough, which can manifest in up to 5% of infants, despite their timely receipt of the anti-HBV birth-dose vaccine [9,10,11]. In addition, vaccination cannot help those already infected. Eradication of infection requires complementing vaccination with the effective treatment of those already infected. The World Health Organization and its partners came up with the Global Health Sector Strategy (GHSS) on Viral Hepatitis, a key objective being to reduce new infections by 90% and mortality by 65% compared to the baseline year of 2015 [12]. To achieve this requires real-time information for focused action and timely intervention. This emphasizes the need for proactive case-search, enhanced clinical care, and expanded availability of HBV vaccination. Furthermore, a thorough understanding of the virus biology and its interactions with the host will guide the development of better treatment strategies. Our review outlines the existing knowledge on HBV biology, the complex interplay with other infections, and existing gaps in knowledge that could accelerate control efforts.

## 2. HBV Burden

About thirty million people are newly infected with HBV every year in the world and cumulatively 296 million persons currently live with chronic HBV. Of these, 2.7 million are coinfected with Human Immunodeficiency Virus (HIV) [13]. Although every year an estimated 1 million HBV related deaths occur worldwide, less than 10% of infected people are diagnosed and only about 1% are able to access treatment [14].

The disease burden is disproportionally distributed with Africa and Asia Pacific together contributing 68% of the global burden [15]. This could be attributed to the lax vaccination policies and the absence of continuous surveillance in the most burdened areas of Africa [16]. Within Africa alone, more than 60 million people are infected with hepatitis B, which amounts to over 60,000 deaths annually [17]. Most of these infections occur early in childhood, either through vertical transmission from a hepatitis B-infected mother to their child or by horizontal transmission during the first five years of life from contact with infected family members or close friends [18].

Candotti et al., in 2006 [19], reported that despite the high endemicity, little was known about the HBV-genotype distribution across Africa until recently. In Ghana, HBV genotype E is prevalent (87%) among blood donors while genotypes A and D constitute minor groups of strains accounting for 10% and 3%, respectively. The identification of one mixed infection of genotypes A and E confirmed the occurrence of HBV co-infection or superinfection with multiple strains. The predominance of genotype E is consistent with recent data obtained from neighbouring countries and most of Western Africa, from Senegal to Angola.

In Ghana, Abesig et al., in 2020, estimated the HBV burden with meta data from seroprevalence studies published between 2015 and 2019 to be 8.36% in adults, 14.30% in adolescents, and 0.55% in children under five years (10). In a recent multi-centre, cross-sectional study, which reviewed registers in 22 health centres across Ghana, a pooled HBV prevalence rate of 11.4% was estimated, making Ghana a hyper-endemic country [20]. The burden was disproportionately distributed within the country, with the savannah region having the highest (22.7%) and the Greater Accra region recording the least (6.4%) seroprevalence rates. A comprehensive surveillance system is needed to assess the community burden and accurately estimate the true burden of the disease in Ghana. Moreover, continuous active molecular surveillance is needed for early detection, tracking transmission, and monitoring viral evolution. This could be the basis for formulating policies, making a case for the implementation of birth-dose HBV vaccination as well as routine adult vaccination to strengthen control efforts within the country. Furthermore, it will aid clinicians in patient management through adopting better treatment measures.

## 3. Overview of Viral Biology

### 3.1. Virus Structure and Genome

The Hepatitis B virus (HBV) is the prototype virus of the *Hepadnaviridae* family of viruses [21]. Members of this family are hepatotropic DNA viruses known to infect birds (avihepadnavirus) and mammals (orthohepadnaviruses). Additionally, fish and amphibian *Hepadnaviruses* have been reported recently [22,23]. These viruses share similarities in their genome organisation as well as their replication approach, with up to 40% and 20% sequence diversity amongst orthohepadnaviruses and avihepadnaviruses, respectively [24].

Three types of HBV-virion particles are usually observed in the serum of infected persons. The infectious virion, also known as the Dane particle, is 42–45 nm in diameter, made up of HBsAg embedded in a lipid envelope, encasing the viral nucleocapsid containing a reverse transcriptase tethered to the nucleic-acid material [25,26]. The other two are subviral particles (22–24 nm), filamentous and spherical in shape, both comprising HBsAg embedded in a host-derived lipid membrane but lacking viral DNA [27]. Interestingly, the subviral particles outnumber the infectious particles by 100 to 100,000-fold in the blood and play immunomodulatory and immunoinhibitory roles. Their role has been extensively reviewed elsewhere [28,29].

The viral nucleic-acid material is a circular, partially double-stranded DNA of approximately 3.2 kb in size that is organized into four overlapping open reading frames (ORFs), each with its own promoter yet sharing a single polyadenylation sequence [26]. The ORFs encode the surface antigen (HBsAg), core antigen (HBcAg), polymerase, and X (HBxAg) proteins. Due to their small genome size, viral proteins are encoded in overlapping translational frames with regulatory elements lying within protein-coding sequences [30]. Additionally, the HBV virion has a highly error-prone polymerase that increases its mutation rate to about 2.0 × 10^−5^ nucleotide substitutions per site per year [31]. Looking at the small genome size, the overlapping nature of ORFs, and the significance of each nucleotide position, such a high mutation rate could be detrimental to the survival of the virus. Remarkably, HBV can maintain replication competence and survive amidst the potential of genome rearrangement.

The virus is divided into four major serotypes (adr, adw, ayr, and ayw) based on antigenic epitopes presented on its envelope proteins, and into eight major genotypes (A–H). Differences between genotypes affect the disease severity and course, the likelihood of complications, the response to treatment, and possibly vaccination. There are at least 10 HBV genotypes, named A to J, classified based on a sequence divergence of more than 8%, as well as multiple sub-genotypes with sequences varying by 4–8% [32]. These exhibit a unique geographical distribution (Table 1) and may also show distinct natural courses of infection [31]. Additionally, some distinct routes of infection have been observed to be common in areas with specific genotypes [32,33,34]. In natural infections with HBV, there is an excess of empty non-infectious subviral particles (SVP) that do not contain the viral capsid. They can contain all three forms of the HBV envelope proteins: L, M, and S. These share a common C terminus, with M containing the pre-S2 domain relative to S and L containing the pre-S1 domain relative to M (15). There is good evidence that, during infection, a domain within the pre-S1 of L is what interacts with an as-yet-unidentified host receptor(s).

The interplay between viral proteins and the innate responses remains central to the development of curative treatment strategies against HBV and HDV. Experimental infection systems and patient analyses support the notion that HBV avoids innate immune recognition but that the co-infection with HDV appears to cause profound changes in the infected liver. Dandri M. et al., in 2021 [35], observed higher production of chemokines and inflammatory cytokines, as well as the increased antigen-presentation capabilities determined in HBV/HDV infection, boosts the ability of immune cells to recognise infected cells, augments liver inflammation, and accelerates pathogenesis. Through research, the role of distinct HBV and HDV proteins in modulating the antiviral responses in infected hepatocytes has gained recognition and highlighted the importance of viral activity in counteracting the first line of host defences.

Geographical distribution of the various HBV genotypes is summarised in Table 1. Nevertheless, there is still much to be understood about HBV genomic features and their role in establishing and maintaining infection. More studies on the molecular epidemiology of HBV are required to assess their impact on disease progression and response to treatment, especially in hyperendemic regions. For instance, in a recent article, Lemoine et al., in 2023 [36], maintains that the status quo in the management of hepatitis B in Africa is intolerable and they indicated that, to prevent new infections, Birth Dose Vaccine (BDV) must be made available for all new-borns in Africa. The authors reiterated that the current model that involves repeated laboratory assessments and complex criteria for the initiation of antiviral therapy, together with the cost of diagnostics, is an obstacle to those requiring therapy. However, they observed that there is a lack of education and awareness, poor access to relevant laboratory tests and imaging, and inconsistent access to appropriate medication. This is particularly pertinent in resource-constrained settings. Decentralisation, integration of services, and task-sharing, with adequate funding of the required infrastructure, will be key to upscaling delivery of care. However, new approaches such ‘Treat all’ or ‘Treat-all-except’ approaches do need further evidence from implementation-science research, including qualitative evaluation of understanding, attitudes, and acceptability of such approaches by both patients/carriers and policy makers.

Furthermore, there is a need for in-depth current studies on viral biology, mainly for the purposes of coming up with better management strategies for infected people and more effective preventative measures for the rest of the population. Intensified surveillance systems in endemic regions will also aid monitoring efforts to identify new genomic recombination events and their impact on control strategies.

### 3.2. Pathogenesis

HBV can be acquired through two main routes—perinatally, from infected mothers to their new-borns, which accounts for a majority of cases worldwide, and horizontal transmission through contact with an infected person’s body fluids, equipment for body piercing, tattoos, and injecting-drug use. Generally, the mechanisms by which HBV accesses and gains entry into hepatocytes is not fully understood. [37]. The heparan sulphate proteoglycans (HSG’s), sodium taurocholate co-transporting peptide (NTCP), and the epidermal growth factor receptor (EGFR) are some of the receptors that mediate this internalization although there are likely to be others [38,39]. Both the NTCP and HSG’s are hepatocyte-specific receptors, thus explaining the virus’ hepatotropic nature [40]. In-depth studies into these receptor interactions could significantly contribute to finding a cure to HBV through inhibition of viral spread within the liver. Recently, some compound leads have been shown to selectively inhibit the virus-receptor function of NTCP [41].

Following internalization, the virus uncoats and releases its nucleocapsid into the cytoplasm followed by release of the genome into the hepatocyte nucleus [42]. Here, HBV DNA, which is partially double stranded, is repaired by viral polymerase or host repair mechanisms, giving rise to a covalently closed circular DNA (cccDNA) that serves as the template for the synthesis of sub-genomic and pre-genomic RNA (sgRNA & pgRNA) [43]. A nucleocapsid encloses the viral DNA and a DNA polymerase that has reverse-transcriptase activity. The outer envelope contains embedded proteins that are involved in the viral binding of, and entry into, susceptible cells. The virus is one of the smallest enveloped animal viruses. The 42 nm virions, which are capable of infecting liver cells, are referred to as “Dane particles”. In addition to the Dane particles, filamentous and spherical bodies lacking a core can be found in the serum of infected individuals. These particles are not infectious and are composed of the lipid and protein that forms part of the surface of the virion, the surface antigens (HBsAg), and is produced in excess during the life cycle of the virus. The cccDNA is highly stable and can remain in hepatocytes indefinitely, becoming a permanent template for sg/pg RNA synthesis, thus promoting viral persistence in the cells. The sgRNA is translated into viral proteins whereas the pgRNA together with the polymerase is packaged to form the nucleocapsid where pgRNA is reverse transcribed into negative-stranded DNA, followed by the generation of rcDNA from plus-strand synthesis [44]. The nucleocapsids are either imported back into the nucleus for cccDNA synthesis or packaged and secreted through the endoplasmic reticulum (ER) to infect other hepatocytes.

As a result, HBV exits in the infected cells without causing cytocidal changes within the hepatocyte, although other cellular changes that occur tend to increase a person’s risk of liver injury and hepatocellular carcinoma [45]. Additionally, viral variants and components seem to play a role in the severity of damage that occurs in the liver [46]. The main culprit for liver damage in HBV infection is said to be the immune system, where the HBV-specific T-cell response initiated for viral clearance is dysregulated. Both innate and adaptive immune arms play significant roles in liver pathology and viral clearance. Not much is known about innate immune response in HBV infection and, previously, it was thought that HBV did not elicit an innate immune response [47]. However, some studies have suggested a role for polymorphonuclear cells in HBV-related liver inflammation, which may be achieved through cytokine production and recruitment of other immune cells. Furthermore, stress responses such as release of reactive oxygen species, endoplasmic reticulum (ER), and mitochondrial dysregulation may initiate cell-death signals in infected hepatocytes [48,49]. Gherlan GS, in 2022 observed that the factors that determine the outcome of occult hepatitis B infection (OBI) are now better understood, with host factors (immune or epigenetic) being identified as seemingly the main contributors. He stated that viral factors are important but account for only a minority of OBIs, but some external factors can contribute to its appearance by interfering either with the host immune system or with the lifecycle of HBV. Of these, HIV and HCV co-infections are notable.

In acute infection, viral clearance is achieved through the work of cytotoxic CD8+ T cells and helper CD4+ T cells, which help to eliminate infected cells and stimulate antibody production against viral antigens respectively [50]. Much remains to be studied about the role of these immune cells since, although we know their function is impaired in chronic infection, the mechanisms of impairment and the true significance is not clearly understood. A clear understanding of the HBV immune-response process and its role in cell damage can be exploited for management.

**Table 1 viruses-16-00724-t001:** Overlap of HBV and HDV genogroups and associated outcomes of infection.

HBV Genotype	Dominant Region	HDV Genotype Dominant in the Region	Reported Outcome of HDV Infection
A	North-western Europe Northern America Central Africa	HDV-1	Varied severity of liver disease
B	Asia	HDV-2, HDV-4	Comparatively less severe clinical manifestation
C	Asia	HDV-2, HDV-4	Comparatively less severe clinical manifestation
D	Worldwide	HDV-1	Varied severity of liver disease
E	Sub-Saharan Africa	HDV-5 to 8	Yet to be clarified
F	South & Central America	HDV-3	Most severe form of HDV
G	France & USA	HDV-3	Severe fulminant Hepatitis
H	South America	HDV-3	Severe fulminant Hepatitis
I	South America	HDV-3	Severe fulminant Hepatitis
J	Japan	HDV-2, HDV-4	Comparatively less severe clinical manifestation

References: [28,30,44].

## 4. Treatment

Acute hepatitis B infection does not usually require treatment and most adults clear the infection spontaneously [51]. Early antiviral treatment may be required in fewer than 1% of people, that is the fulminant or the immunocompromised individuals. Treatment of chronically infected persons with persistently elevated serum alanine aminotransferase, a marker of liver damage, may be necessary to reduce the risk of cirrhosis and liver cancer [52]. Treatment lasts from six months to one year, depending on medication and genotype [53].

Although none of the available medications can clear the infection, they can stop the virus from replicating, thus minimising liver damage. The licensed medications for treatment include antiviral medications lamivudine, adefovir, tenofovir disoproxil, tenofovir alafenamide, telbivudine, entecavir, and the two immune system modulators interferon alpha-2a and PEGylated interferon alpha-2a. In 2015, the World Health Organization recommended tenofovir or entecavir as a first-line agent [54]. Those with current cirrhosis are in most need of treatment [54].

Ultimately, HBV elimination can be defined by complete suppression of HBV DNA levels, the loss of HBsAg, and seroconversion to anti-HBs antibodies after stopping antiviral therapy. Loss of HBsAg levels is critical since HBsAg levels are surrogate markers for levels of transcriptionally active covalently closed circular DNA (cccDNA), meaning that if HBsAg is eliminated, the virus is most likely inactivated [53]. In chronic-HBV individuals who are negative for HBeAg, the bulk of the HBsAg is produced from integrated HBV DNA. Although adaptive immunity is key to controlling and clearing HBV infection, the role of innate immunity cannot be ignored. Adaptive immunity depends on the activation signals and cytokines secreted by the innate immune system. HBsAb can bind to HBsAg to limit its spread and kill or phagocytose HBV-infected cells.

It seems unlikely that the disease will be eliminated by 2030, the goal set in 2016 by WHO. However, progress is being made in developing therapeutic treatments. In 2010, the Hepatitis B Foundation reported that three preclinical and 11 clinical-stage drugs were under development, based on largely similar mechanisms. In 2020, they reported that there were 17 preclinical- and 32 clinical-stage drugs under development, using diverse mechanisms [55,56].

### Comorbidities with Other Liver Pathogens

Other microbes may co-infect the liver together with HBV, such as hepatitis A, C, D, and E viruses. Additionally, HBV can co-occur with non-communicable liver conditions including steatohepatitis and alcoholic liver disease. Although parasites such as liver flukes and biliary-tract bacterial infections also affect the liver, these are relatively less common [57]. *Plasmodium* species, the causative agent for malaria, asexually reproduces in the human liver as part of its life cycle. Its activities in the liver have not been linked to significant hepatocyte damage, yet immune response to this parasite in the liver may potentially influence that of HBV [58,59]. Moreover, the few studies on HBV-malaria coinfection have reported varied levels of interaction between the two pathogens [60,61]. Here, we examine the impact of other liver infections on HBV outcome.

Hepatitis Delta Virus (HDV): According to the WHO, about 5% of all HBV-infected people also have HDV [62], however Chen, Shen [63] estimate this to be 10.58% and this number is bound to increase with HBV incidence yearly. On its own, the risk of HBV-induced liver cirrhosis ranges from 6% in America to up to 38% in sub-Saharan Africa and 39% in East Asia [64]. However, compounded with HDV, the risk could be significantly higher.

The Hepatitis D virus, which was first discovered in 1977 among HBV patients with severe liver damage, is the smallest human-infecting RNA virus, with 36–40 nm diameter and roughly 1.7 kb single-stranded negative-sense circular RNA [65,66]. Known as defective because of its inability to establish an infection on its own, the virus is spherical and made up of an outer lipoprotein envelope, composed of HBsAg, which encloses a ribonucleoprotein. Not only does the HBsAg structure in HDV bear semblance to spherical SVPs more closely than filament SVPs and even Dane particles, their assembly and cell exit utilise the same pathways [28,67] The HDV ribonucleoprotein is unconventional, comprising genomic RNA complexed with 70–220 HDV-specific antigens known as the delta antigens (HDAg) [68]. The HDAg exists in two isoforms: L-HDAg, which has 19 additional amino acids at the C-terminal compared to S-HDAg [69]. These two isoforms have distinct roles in the HDV life cycle, with S-HDAg being essential for replication, while L-HDAg is crucial for assembly due to its C-terminal 19 amino acids, which contain the virus-assembly signal [70,71]. HDV infection could be classified either as a co-infection with HBV, where both viruses are acquired simultaneously, or as a superinfection, where a chronic HBV patient later acquires HDV [72].

The mechanism of viral entry into hepatocytes is like that of HBV. The virus gains entry into cells by interacting with the NTCP and HSGs on the hepatocyte surface [64]. Despite having an HBV-dependent entry pathway, HDV viral replication and assembly are distinct from HBV [73]. Additionally, HBV DNA suppression among superinfected persons has been observed with controversy on its association with faster progression to hepatocellular carcinoma (HCC). This could be due to the dominant HDV genotype in the region, some of which yield severe hepatitis than others (Table 1). Since its identification as the cell surface receptor, NTCP, for HBV and HDV entry into hepatocytes, the search for molecules interfering with its binding led to the design of bulevirtide (BLV). This large polypeptide mimics a region of the pre-S1 HBsAg and blocks viral entry by inhibitory competition. BLV was initially tested in cell cultures, animal models, and, more recently, in Phase I–III human trials. As a monotherapy or in combination with peginterferon, BLV is well tolerated and exhibits potent antiviral activity. Plasma viremia significantly declines and/or becomes undetectable in more than 75% of patients treated for >24 weeks with BLV. However, serum HBsAg concentrations remain unchanged with BLV treatment even though plasma viremia drops. No selection of BLV resistance in HBV/HDV has been reported in vivo to date.

In the cell, HDV uncoats and its RNA genome is translocated to the cell’s nucleus, with the help of HDAg, where host RNA polymerase I and II are hijacked to make copies [74]. Looking at the virus structure, it is not surprising that HDV lacks the ability to replicate independently, relying on host RNA polymerase and ribosomes to make new copies of viral RNA and proteins respectively. Replication of HDV occurs by the rolling circle mechanism where multimeric liver transcripts complementary to the HDV genome, known as the antigenome RNA, are transcribed then self-cleaved by intrinsic ribozyme activity to separate monomers from multimeric transcripts. These are joined together to create a circular antigenomic template for the synthesis of the viral genomic strand [75]. This is then packaged into HBsAg envelopes and released in a clathrin-mediated manner [76]

Currently, there are at least eight HDV genotypes, named genotypes 1 to 8, and these show geographical restrictions just like in HBV (Table 1). Genotype 1 has a global distribution and is most prevalent in Europe, North America, and parts of Asia, with a variable course of infection; whereas genotype 3, found in the Amazon Basin, is associated with early onset of HCC and acute liver failure [77]. Furthermore, genotype 2, found in Russia, Taiwan, and Japan, is associated with higher rates of remission than genotype 1 and genotype 4 in the same location and is associated with faster progression to cirrhosis [78]. Genotypes 5, 6, 7, and 8 are localised in Africa and African migrants in Europe, with limited data and the course of infection poorly classified [79,80]. The co-localization of these viral genotypes with specific HBV genotypes may explain the varying outcomes of infection and could be exploited to guide treatment strategies since the effectiveness of currently available treatment regimen is genotype dependent.

Large gaps in epidemiological data on HDV prevalence result in underestimated prevalence and incidence rates. For instance, in Ghana and many endemic developing regions, routine screening of HBV infected persons for HDV is lacking and prevalence estimates are based on sporadic screening of groups in various studies [81,82]. Additionally, since the HBV burden may be underestimated, it is likely that the HDV burden is also underestimated. Surveillance studies to accurately estimate prevalence, and studies to clearly define the role of genotypic differences in the severity of liver damage, are needed to guide control strategies. Global eradication of HDV is therefore directly linked with the eradication of HBV.

***Plasmodium* liver stage infection:** *Plasmodium* is the protozoan parasite responsible for malaria infections; it caused 229 million cases in 2019 with about 400,000 deaths [82]. The majority of malaria cases occur in Africa and, in 2019, accounted for 95% of global cases. The parasite is spread through an infectious bite from a female *Anopheles* mosquito and, currently, there are five species known to infect man, including *P. falciparum*, *P. vivax*, *P. ovale*, *P. malariae*, and *P. Knowlesi*, of which *P. falciparum* is accountable for the majority of deaths [83].

The human parasite’s life cycle takes place within two hosts—the mosquito and man. The mosquito injects the parasite in its sporozoite form into the skin of a human host, where it finds its way into the bloodstream [84]. The parasite travels to the liver and develops into schizonts which in turn mature and rapture, releasing merozoites into the bloodstream to invade red blood cells and give rise to the clinical symptoms of malaria [85]. Although clinical symptoms are often not obvious at the liver stage, hepatic dysregulation has been reported in severe malaria characterized by hepatocyte necrosis, granulomatous lesions, Kupffer cell hyperplasia, and malarial pigmentation among others [86]. In asymptomatic infection, abnormal total bilirubin levels have been observed but resolved after a few days [87]. Elevated liver enzyme levels have also been reported in uncomplicated malaria at the time patients reported and was associated with parasite load, pointing to *Plasmodium* involvement rather than a drug-induced effect [88]. While the exact cause of these abnormalities is unclear, it could influence the immune response to other infections and their pathology.

In sub-Saharan Africa where HBV is endemic, *Plasmodium* infections are also endemic and may co-occur in individuals. Studies on HBV and *Plasmodium* coinfection are few and sporadic, making prevalence estimations challenging. However, a 6% pooled prevalence rate has been reported, with places like Gambia and Nigeria reporting 10% and 7%, respectively [57]. It has been established that areas endemic for malaria and HBV infection largely overlap geographically. A recent study has suggested the existence of an interaction between the two pathogens in symptomatic co-infected individuals on the South American continent. However, data presented by Freimanis GL et al., in 2012 [80] suggest that, in sub-Saharan Africa, asymptomatic co-infections with these two ubiquitous pathogens do not appear to significantly affect each other and evolve independently. Whereas HBV infections tend to be lifelong, *Plasmodium* infections are usually short-term and resolve with treatment in weeks. The *Plasmodium* liver phase is also short-lived, with the covert immune response characterised by the induction of immune inhibitory pathways shortly after the inflammatory response is mounted [89,90]. With the strong impact HBV and host diversity has on infection outcome, it is possible that immune response to *Plasmodium* infections and its impact on liver health could be modulated by the presence of HBV and vice versa. Also, the chronicity of HBV infection, and the potential of acquiring *Plasmodium* infections several times during HBV infection, may influence infection outcome of both diseases. This could also affect liver integrity in the long run, making it imperative to study and understand.

**Hepatitis C virus infection**: HCV is a single-stranded RNA virus which belongs to the Flaviviridae virus family and the Hepacivirus genus. An estimated 58 million people live with HCV worldwide, of which 3.2 million are adolescents and children [13]. Furthermore, 1.5 million new infections are reported each year with 290,000 deaths mainly through end-stage liver disease. Because of shared routes of transmission, HBV and HCV coinfection is common and can be seen in up to 30% of chronic-HBV-infected persons and about 10% of HCV-infected persons [91,92].

The HCV genome is 9.6 kb in length, organised into one continuous open reading frame (ORF) flanked by highly structured UTRs at both the 5′ and 3′ ends [93]. This ORF encodes a 3010 amino acid long polyprotein which goes through post translational modification to make three structural proteins, Core (C) and Envelope1 and 2 (E1 and E2); and 7 non-structural proteins, NS2, NS3, NS4A, NS4B, NS5A, NS5B, and p7 [94]. Like HBV, HCV replicates mainly in hepatocytes and, although their nucleic acid material differs, both at some point in replication yield an RNA intermediate that theoretically could interact [95]. There are conflicting reports on HBV/HCV interaction in vivo. Some studies suggest that both viruses can replicate in the same cell without restriction while others report a mutual suppression of each virus’ replication [96,97,98].

When HBV and HCV infections occur separately, they each contribute to liver damage by provoking an excessive inflammatory response against the respective viruses. Nevertheless, when both viruses infect an individual simultaneously, there are conflicting reports regarding the resulting liver disease outcomes. Some studies suggest the simultaneous suppression of both viruses, while others indicate the progression of one virus to a chronic state. In certain cases, fulminant hepatitis has been reported in dual infections [99,100]. These discrepancies underscore the potential role of host factors in shaping the divergent disease outcomes observed. Furthermore, certain studies have documented enduring epigenetic alterations in HCV, which remain present even after treatment and recovery, potentially increasing the risk of hepatocellular carcinoma (HCC) later in life [101,102]. HCV might become the first curable chronic disease due to the remarkable efficacy of the newly introduced direct-acting antiviral drugs (DAAs). Interferon-free regimens, based on combinations of DAAs with pan-genotypic activity, allow for shorter courses of treatment without severe side effects. However, the high cost of the DAAs precludes universal replacement of the suboptimal interferon-based therapy for chronic hepatitis C. Across the 9.6 kb genome of HCV, several regions have been extensively analysed in relation to treatment outcome, but the core region that is mostly used for HCV genotyping and classification has been reported to antagonize the antiviral response induced by IFN by interacting with the IFN-activating and -signalling pathways. Sultana C. et al., in 2016 [103] confirmed core substitutions are also found in Caucasian patients and, together with age and IL28B genotype, can be used as predictors of the outcome of interferon-based therapy. The study concluded that HCV core mutations can help distinguish between patients who can still benefit from the affordable IFN-based therapy and those who must be treated with DAAs to prevent the evolution towards end-stage liver disease [103]. While effective treatment can cure HCV, its lingering impacts may persist and potentially lead to cancer in the future. Thus, people who are cured of HCV could still experience the dual impact of HCV-HBV coinfection if they later acquire HBV.

**Human Immunodeficiency Virus (HIV)**: Human immunodeficiency virus (HIV), the causative agent of acquired immunodeficiency syndrome (AIDS), remains a significant cause for public-health concern worldwide. As at 2021, an estimated 38 million people live with HIV infection, over a million new cases were recorded, and 650,000 people died from it [97]. While a definitive cure for the disease remains elusive, antiretroviral drugs (ARVs) have played a crucial role in extending the life expectancy of individuals living with HIV, bringing it closer to that of uninfected individuals. ARVs employ diverse mechanisms to hinder viral replication, thereby alleviating the strain on the immune system and reducing the vulnerability to opportunistic infections. By virtue of their action, ARVs also contribute to a reduction in HIV transmission rates and a substantial enhancement of the overall quality of life for those affected by the virus.

Globally, 2.7 million people living with HIV (PLHIV) also have HBV (7.6% prevalence), with a majority of these in sub–Saharan Africa (1.9 million) where HBV is endemic [104]. Certainly, the ongoing lifelong management of HIV introduces the potential for hepatotoxicity, and when compounded with HBV infection, which is recognized for its liver-related complications, the consequences could be dire. Moreover, liver-related mortality amongst HIV-HBV-coinfected people is said to be 17x higher than in HBV-mono-infected people [97]. HIV infection has myriad effects on adipocyte biology that might co-ordinately impact liver disease. The most obvious connection is with the accumulation of additional liver fat (steatosis), which in some instances also is associated with disease (inflammation or fibrosis). Rosca A, et al., and Nguyen MH et al., both in 2020 [105,106], respectively wrote on the ‘Liver function in a cohort of young HIV-HBV co-infected patients on long-term combined antiretroviral therapy’ and ‘Hepatitis B Virus: Advances in Prevention, Diagnosis, and Therapy’.

Several options exist for the treatment of hepatitis B including interferon, pegylated interferon, lamivudine, adefovir, entecavir, and telbivudine, as well as tenofovir, which has been licensed. The antivirals can be divided into “lamivudine-like” and “adefovir-like”, which clinically differ and their resistance profiles make them good combination partners, even in the absence of synergy in antiviral potency. The “adefovir-like” drugs best used in practice are adefovir in the HIV-infected patient in need of anti-HBV therapy while not yet needing anti-HIV therapy [104]. In other patients, tenofovir is to take over where adefovir is currently used, given its lower toxicity and higher activity. It is probable that all could be well combined with lamivudine, which will soon be off patent. Thus, it might be a cheap but potentially very active addition to any “adefovir-like” drug, given their different resistance profiles. However, in the case of tenofovir, this is not required, given its existence in combination with the lamivudine-like drug emtricitabine.

Aside from liver complications, secondary HIV infection in HBV-infected adults has been shown to increase the risk of HBV progressing to chronicity six-fold [98]. Moreover, the progression of HBV infection is notably affected, particularly in terms of changes in HBV antigen and antibody expression, along with an elevated susceptibility to HDV [99].

## 5. Control Strategies for HBV Infection

Presently, there is no cure for HBV infection and the treatment options available function mainly to reduce viral load so that liver damage can be slowed to the barest minimum. Thus, chronic HBV is a lifelong infection although, in acute cases, the immune system may adequately clear the infection completely. Fortunately, there exists a highly effective vaccine against HBV, which has played a pivotal role in the strategies of global control programs since its development.

The WHO’s Global Health Sector Strategy (GHSS) on viral hepatitis hopes to achieve elimination of viral hepatitis by 2030. This means reducing the annual disease incidence and mortality by 90% and 65%, respectively, using 2015 data as baseline [12]. To achieve this, control programmes have been encouraged to pursue HBV and HCV elimination simultaneously, although a country may prioritise based on their peculiar situation. With respect to reducing HBV incidence and mortality, vaccination must be complemented with a cure. The definition of a cure itself is a source of controversy, since, clinically, a cure means moving a chronic-HBV-infected person with risk of liver disease to the state of an uninfected person. Due to the latent persistence of HBV cccDNA in infected hepatocytes, this is difficult to achieve and may even require lifelong treatment. Thus, the realistic aim is to achieve a functional cure, meaning to maintain reduced viral load as well as other viral markers in the blood after therapy ceases.

The main goals of treating chronically infected HBV patients are to improve survivability by limiting progression to HCC, limiting mother-to-child transmission in pregnant women, and preventing extrahepatic complications. The possibility of achieving these goals depends on factors including the stage of infection and patient’s age when therapy is initiated. Currently, available treatment options for HBV are nucleo(s)tide analogs (NUCs) and interferon-based therapy. NUCs commonly approved for HBV treatment include Tenofovir disoproxil, Lamivudine, and Entecavir. These drugs function by inhibiting viral replication [107]. The activity of these NUCs can significantly reduce HBV DNA levels, but they are ineffective against cccDNA, which maintains the chronic HBV state [108]. It has been demonstrated that individuals with significant HBsAg decline have a commensurate loss of infected cells with transcriptionally active cccDNA, while individuals without HBsAg decline have stable or increasing numbers of cells producing HBsAg from invertebrate-derived DNA (iDNA). While NUC therapy may be effective at controlling cccDNA replication and transcription, innovative treatments are required to address iDNA transcription that sustains HBsAg production. Also, interferon-based therapies are more effective for certain HBV genotypes than others [109]. Up until 2020, no specific treatment existed for HDV even though drug discovery for HDV-specific antivirals is ongoing [110]. In 2020, Bulevertide (BLV) received approval from the European Medicines Agency (EMA) for the treatment of HDV. BLV impedes the attachment of HBsAg to NTCP and has demonstrated synergistic effects with Peg-IFNα, yielding superior outcomes compared to monotherapy with either drug [111]. Developing effective therapy for HDV and complementing HBV vaccination with effective management of chronic patients could significantly reduce the burden of infection and lead towards HBV elimination.

## 6. Conclusions

The observations we made indicate a need for prevention and control of, generally, serum hepatitis in hyperendemic and low-resourced countries, especially in the West African sub-region. There is the need for operative strategies which requires comprehensive investments to interrupt the transmission of serum hepatitis and reduce the consequential morbidity and mortality. The importance of expanding research in the field of HBV cannot be overstated. There is a pressing need to elevate efforts in HBV research to precisely assess prevalence rates, identify at-risk populations, establish treatment priorities, and deepen our comprehension of host-pathogen interactions that could ultimately lead to a cure. Such insights are essential not only for shaping control programs but also for informing the adoption of effective strategies and the development of treatment policies. Since the cure rates are minimal, therapies need to be accessible and affordable since they are required by patients indefinitely. Antiviral treatment with either pegylated interferon or a nucleos(t)ide analogue (lamivudine, adefovir, entecavir, tenofovir disoproxil, or tenofovir alafenamide) should be offered to patients with chronic HBV infection and liver inflammation in an effort to reduce the progression of liver disease. Nucleos(t)ide analogues should be considered as a first-line therapy. Considering the numerous co-infections that have the potential to complicate HBV pathogenesis and control, it is imperative to conduct thorough investigations into their impact on treatment and to formulate improved guidelines for their effective management.

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
