# Peer review of "Hepatitis B Virus Infection: A Mini Review"

_viruses, 2024, doi:10.3390/v16050724_

Round 1

Reviewer 1 Report

Comments and Suggestions for Authors

-

Comments on the Quality of English Language

-

Author Response

Kindly find attached point-by-point responses to the comments from reviewer 1 in addition to the revised manuscript.

Reviewer 2 Report

Comments and Suggestions for Authors

The manuscript by Asandem et al. is not as indicated in the title a review of HBV but an HBV introduction to describe in some details HBV co-infections. Therefore, either the text should expand the HBV part in describing in some detail the importance of genotypes and the critical issue of occult HBV infection or the title should be HBV co-infections. In particular, there is a hypertrophied section on HDV and a prominent place on Table 1 that is not justified in such a short review.

More specifically:

In the introduction lines 23-25, there should be more quotations of articles reporting HBV vaccination failures such as declining protection over time and infectious sexual contact  past age 15 and vaccination at 6 weeks rather than at birth.

In section 2, there should be mention of epidemiological differences according to genotypes and indicate that in Ghana and west Africa, the highest prevalence of CHB is related to genotype E, the most infectious genotype. (Candotti et al. J Viral Hepat 2006; 13: 715-24)

In 3.1 lines 109-112, the authors should mention the article by Lemoine M et al (Lancet Glob Health. 2016;4:e559-67) delineating a strategy for eradication of HBV in west Africa. 

At the end of 3.2, the authors should include a section describing the issues raised by occult HBV infection (OBI) exemplifying the interaction between viral replication and immune response and the object of a massive literature that should not be ignored. In immunological terms, the difference in response between sexes should be mentioned and some articles quoted such as: Allain JP, et al. Hepatitis B Virus Chronic Infection in Blood Donors from Asian and African High or Medium Prevalence Areas: Comparison According to Sex. Viruses 2022: 14: 67381.

In section 4.0, the authors should quote a paper preceding those they quote on HBV and malaria reporting data from Ghana on the subject: Freimanis GL et al.. Hepatitis B virus infection does not significantly influence Plasmodium parasite density in asymptomatic infection in Ghanaian transfusion recipients. PLOs One 2012; 7: e49967.

In the section co-infection with HIV, there should be mention that several ART drugs (Adefovir, Tenofovir) are highly active against HBV replication and can be obtained cheaply to treat CHB.

Comments on the Quality of English Language

English is good, there are a few typos to correct.

Author Response

Kindly find attached point-by-point responses to the comments from reviewer 2 in addition to the revised manuscript

Reviewer 3 Report

Comments and Suggestions for Authors

I have read and reviewed the paper “Hepatitis B virus infection – A mini review” by Diana Asema Asande, Selorm Philip Segbefia Kwadwo Asamoah Kusi, Joseph Humphrey Kofi Bonney - very carefully and observed that the results of the paper are interesting, well organized, the novelty of the results is good and the presentation of the paper is suitable.

I recommend the paper for publication in this journal but after some corrections. My suggestions are as follows:

1.      There are some small grammatical mistakes spread throughout the manuscript that must be fixed.

2.      The abstract should be concise and informative and the Introduction should make a more convincing case for the reasons the study is useful, and clearly state its novelty.

3.      In the chapter 3.1, when describing the role of the subviral particles, please add the capacity of inducing the immunopathogenic effects during HBV infection and the possibility of acting as a helper virus for HDV – as you will have a chapter about HBV-HDV coinfection later in the manuscript

4.      The chapter related to HBV-Human immunodeficiency virus (HIV) coinfection should include a small paragraph about the consequences of HIV on liver function and recent literature studies regarding the current study topic. I suggest the following works to the author:

Rosca A, et al. Liver function in a cohort of young HIV-HBV co-infected patients on long-term combined antiretroviral therapy. Farmacia, 2020, Vol 68, Nr 1, pp 42-47, ISSN: 0014-8237

Nguyen MH, et al. Hepatitis B Virus: Advances in Prevention, Diagnosis, and Therapy. Clin Microbiol Rev. 2020 Feb 26;33(2):e00046-19. doi: 10.1128/CMR.00046-19.

5.      The chapter related to HBV- hepatitis C virus (HCV) coinfection can contain a small explanation about influence of the HCV genomic mutations on the evolution of liver disease. I suggest the following articles:

Sultana C, et al. Impact of hepatitis C virus core mutations on the response to interferon-based treatment in chronic hepatitis C. World J Gastroenterol. 2016 Oct 7;22(37):8406-8413. doi: 10.3748/wjg.v22.i37.8406.

Tsukiyama-Kohara K, et al. Hepatitis C Virus: Viral Quasispecies and Genotypes. Int J Mol Sci. 2017 Dec 22;19(1):23. doi: 10.3390/ijms19010023.

6.      In the conclusion part the significant outcome of the study needs to be emphasized

Comments on the Quality of English Language

Small mistakes, revision is required

Author Response

Kindly find attached point-by-point responses to the comments from reviewer 3 in addition to the revised version of the manuscript.

Round 2

Reviewer 1 Report

Comments and Suggestions for Authors

The efforts of the authors to respond to this reviewers' questions are appreciated and the manuscript is suitable for publication with minor but important corrections which are identified in the accompanying PDF.

Author Response

Kindly find attached

Reviewer 2 Report

Comments and Suggestions for Authors

The revised review by Azandem et al. has significantly improved the relatively poor text initially presented. It remains of insufficient quality.

In revised page 3, it is not co-infection but molecular recombination of genotype E and A or E and D. This should be supported by appropriate reference such as: Garmiri P, et l. J Gen Virol 2009; 90: 2442-51.

Several of the references mentioned in the revised text have not been added to the reference list s it should be. Freimanis et al, Lemoine et al, Gherlan et al., Dandri et al., Candotti et al. etc.

Comments on the Quality of English Language

OK

Author Response

Kindly find attached

Reviewer 3 Report

Comments and Suggestions for Authors

All the requests were properly addressed by the authors.

Author Response

All the requests were properly addressed by the authors